# Social context matters: The role of social support and social norms in support for solidarity in healthcare financing

**Marloes A. Meijer** [1]*, **Anne E. M. Brabers**[1], **Judith D. de Jong**[1,2]

**1** Nivel (Netherlands Institute for Health Services Research), Utrecht, The Netherlands, **2** Department of Health Services Research, Faculty of Health, Medicine and Life Sciences, Maastricht University, Maastricht, The Netherlands

* m.meijer@nivel.nl

## Abstract

In many European countries, including the Netherlands, the healthcare system is financed according to the principles of solidarity. It is important, therefore, that public support for solidarity in healthcare financing is sufficient in order to ensure that people remain willing to contribute towards solidarity-based systems. The high willingness to contribute to the healthcare costs of others in the Netherlands suggests that support is generally high. However, there are differences between groups. Previous research has focused on mechanisms at the individual and institutional level to explain these differences. However, people's social context may also play a role. Little research has been conducted into this. To fill this gap, we examined the role of perceived social support and social norms in order to explain differences in the willingness to contribute to other people's healthcare costs. In November 2021, we conducted a survey study in which a questionnaire was sent to a representative sample of 1,500 members of the Dutch Healthcare Consumer Panel. This was returned by 837 panel members (56% response rate). Using logistic regression analysis, we showed that people who perceive higher levels of social support are more willing to contribute to the healthcare costs of others. We also found that the willingness to contribute is higher when someone's social context is more supportive of healthcare systems that are financed according to the principles of solidarity. This effect does not differ between people who perceive low and high levels of social support. Our results suggest that, next to the individual and institutional level, the social context of people has to be taken into consideration in policy and research addressing support for solidarity in healthcare financing.

## Introduction

In many Western European countries, including the Netherlands, the healthcare system is financed according to the principles of solidarity. Solidarity refers to a social cohesion between individuals. In countries with a solidarity-based healthcare system, this cohesion is established by public social insurance arrangements installed by the government [1]. These arrangements provide citizens with financial support in case of illness. In exchange for this support, people are obliged to contribute to the healthcare system [1]. These contributions are, in part, related

**Data Availability Statement:** The minimal anonymized data set necessary to replicate our study findings has been uploaded to DANS EASY

(https://doi.org/10.17026/dans-2b2-kf3a). The data set may only be used under the conditions laid down in the privacy regulations of the Dutch Healthcare Consumer Panel. Therefore, the dataset is available on request. Data requests will be assessed against the privacy regulations. Additionally, the same minimal anonymized data set is available upon request from prof. Judith D. de Jong (j.dejong@nivel.nl), project leader of the Dutch Healthcare Consumer Panel, or the secretary of this panel (consumentenpanel@nivel.nl).

**Funding:** The Dutch Healthcare Consumer Panel is financed by The Dutch Ministry of Health, Welfare and Sport. The funder had no role in study design, data collection and analysis, decision to publish, or preparation of the manuscript.

**Competing interests:** The authors have declared that no competing interests exist.

to income, so that people with higher income levels contribute more (i.e. income solidarity). In addition, they are unrelated to health risks (i.e. risk solidarity). In that way, financial barriers to healthcare are removed [2].

It is important that public support for solidarity in healthcare financing is sufficient as the healthcare system in many countries is solidarity-based. The functioning of healthcare systems depends on the degree to which citizens put their trust in these systems and are willing to contribute towards them [3, 4]. People are generally very supportive of collectively financed healthcare systems [5]. However, there are differences in the level of support between groups. Differences were found both within countries, for example by age, educational level, and health status [6], and across countries [3]. Previous studies have looked into mechanisms at the individual level (micro level) and mechanisms at the level of the healthcare system as a whole (macro level) in order to explain these differences. With regard to the individual level, self-interest and ideological beliefs were found to be the two main mechanisms affecting support for the solidarity-based healthcare system [7, 8]. Regarding the healthcare system as a whole, the type of healthcare system, healthcare coverage, and healthcare expenditures were found to play a role [3, 9].

However, next to mechanisms at the individual and institutional level, the social context of people (meso level) might play a role in differences in support for solidarity in healthcare financing. The social context consists of the web of social relationships surrounding an individual. Among others, family, friends, and co-workers are part of this context [10]. The social context plays an important role in the emergence of attitudes. Attitudes, including those towards solidarity in healthcare financing, are formed, sustained, and changed through social relationships with the people in someone's social context. Attitudes, thus, can not only be attributed to individual preferences. Rather, they are embedded within a larger social environment [11]. Since the social context differs between people, this may form an explanation for differences in attitudes towards solidarity in healthcare financing. Therefore, it is relevant to focus, not only on the individual and institutional level when explaining differences in support for solidarity in healthcare financing, but also on the influence of people's social context. To our knowledge, little research has been conducted into this to date. We have defined support for solidarity in healthcare financing as the willingness to contribute to the healthcare costs of others, and, thus, connected financial consequences to the show of support [12]. We aim to answer the following research question: "How is the social context of people related to their willingness to contribute to the healthcare costs of others?". Our study focuses on the social context as perceived by people. Two aspects of the social context are studied, which will be explained hereafter: social support and social norms.

## Social support

Social support can be described as "*support accessible to an individual through social ties to other individuals, groups, and the larger community*" [13]. It refers to the resources that members comprising someone's social context can provide. Three types of resources can be distinguished: instrumental, informational, and emotional resources [14]. When assessing social support, perceived social support is the most commonly used measure [15]. Perceived social support refers to "*the confidence that adequate support would be available if it was needed*" [16]. It represents an individual's subjective perception of the degree to which their network is available to provide social support in times of need [13]. People who perceive a high level of social support, thus, feel like they can rely on others to support them [17]. Van Oorschot [18] mentions feelings and sentiments as a motivation to contribute to the welfare state. According to this perspective, the degree to which people feel connected to others affects their willingness

to contribute to the common good. It is expected that individuals who feel more connected to others, because they perceive higher levels of social support, are more willing to contribute [17]. When applied to support for solidarity in healthcare financing, this implies that the willingness to contribute to the healthcare costs of others is related to the degree to which people perceive their own social support. It is expected that people who perceive higher levels of social support will be more willing to contribute. Conversely, people who perceive lower levels of social support are expected to be less willing to contribute. This leads to the following hypothesis:

H1: The willingness to contribute to the healthcare costs of others is higher among people who perceive higher levels of social support.

## Social norms

Social norms specify which actions and attitudes are considered appropriate by a group of people and which are considered deviant. Two types of social norms exist: descriptive norms and subjective norms. Descriptive norms refer to the perception of how others act in similar situations, in other words, doing wat others do. Subjective norms refer to the perception of what most people approve or disapprove of, that is, doing what others think one ought to do [19]. Compliance with social norms leads to social approval and rewards, whereas non-compliance leads to social sanctions [20]. People can gain or maintain group acceptance via social norms [21]. Because of this, it is expected that people will develop and express attitudes that are consistent with the social norms of their social context in order to avoid exclusion [22]. Social norms, thus, contribute to conforming to the attitudes, beliefs, and behaviours of others. This leads to a similarity of attitudes within groups [23, 24]. When applying this to support for solidarity in healthcare financing, it is expected that people are more willing to contribute to the healthcare costs of others when their social context is supportive of healthcare systems that are financed according to the principles of solidarity. In contrast, when their social context disapproves of solidarity-based healthcare systems, it is expected that people are less willing to contribute. This leads to the following hypothesis:

H2: The willingness to contribute to the healthcare costs of others is higher among people whose social context holds social norms that support healthcare systems that are financed according to the principles of solidarity.

## The interaction between social support and social norms

Perceived social support and social norms are not only expected to influence, independently, a person's willingness to contribute, but also in interaction with each other. Cullum et al. [25] suggested that the extent to which people attend to social norms depends on the degree to which they feel supported by the members comprising their social context. They argued that people are more likely to adhere to social norms when they do not feel supported [25]. This is because low levels of perceived social support contribute to feelings of being disconnected from others within the social context. To restore this connection, people will adhere to the norms of their social context since this fosters social approval [20, 26]. Applied to support for solidarity in healthcare financing, this implies that the willingness to contribute to the healthcare costs of others is more consistent with the social context's norms regarding support when perceived social support is low. Conversely, when people perceive high levels of social support, their willingness to contribute will be less in line with the norms of their social context. This leads to the following hypothesis:

H3: The willingness to contribute to the healthcare costs of others is more consistent with the norms of the social context for people who perceive lower levels of social support.

## Methods

### Setting

Data were collected using the Dutch Healthcare Consumer Panel, an access panel that is managed by Nivel (the Netherlands Institute for Health Services Research). It collects opinions on, knowledge about, and experiences with healthcare in the Netherlands [27]. At the time of the data collection (November 2021), the panel consisted of approximately 11,000 members from the general Dutch population aged 18 years and older. Several demographic characteristics of the panel members are known such as age, gender, and educational level. The panel can only be joined through invitation. It is not possible for people to sign up on their own initiative. Upon membership, panel members are being informed of the purpose, scope, method, and use of the panel. Based on this information, participants can give permission to participate in the panel. A written informed consent (since 2020 this consent can also be given digital) is obtained at the time of registration of a new member to the panel. Pseudonymised data were analysed and processed in accordance with the privacy policy of the Dutch Healthcare Consumer Panel. The panel complies with the General Data Protection Regulation (GDPR) [28]. According to Dutch legislation, approval by a medical ethics committee is not obligatory for carrying out research using the panel. Participation is voluntary and members are not forced to participate in surveys, or to answer questions within the surveys. They can stop their membership at any time without having to give a reason. Respondents' returning of the questionnaire was considered as consent to participate [29]. Panel members were informed about the subject and length of the questionnaire via the invitation letter. All methods were carried out in accordance with the STROBE guideline.

### Questionnaire

A questionnaire was sent to 1,500 members of the Dutch Healthcare Consumer Panel in November 2021. The sample was representative of the adult Dutch population with regard to age and gender. Representativeness was ensured by matching the age and gender composition of the panel members in the sample to that in the general population, using data from Statistics Netherlands (CBS) [30]. The questionnaire included questions about the willingness to contribute to the healthcare costs of others, perceived social support, and social norms regarding solidarity in healthcare financing. The questionnaire could be filled in online or by post, depending on the personal preference of the panel members. Respondents had four weeks to complete the questionnaire. One postal and three electronic reminders were sent to members who had not yet responded. The questionnaire was returned by 837 panel members (56% response rate). This response rate is similar to that of other studies conducted through the Dutch Healthcare Consumer Panel. In addition, our response rate is in line with average response rates in survey studies reported in the literature [31, 32].

### Measures

**Support for solidarity in healthcare financing.** Support for solidarity in healthcare financing was defined for this study as the willingness to contribute to the healthcare costs of others. This has also been done in other studies on healthcare solidarity [33]. The willingness to contribute was measured by asking respondents the following question: "Are you willing to

pay for healthcare treatments in the basic health insurance that you do not, or not yet, use, but others do?". They could answer this question with No (= 0) or Yes (= 1).

**Perceived social support.**   Perceived social support was measured using the validated ENRICHD Social Support Instrument (ESSI), which we translated into Dutch supported by a translator who is a native English speaker [34, 35]. This instrument assesses the different dimensions of social support (instrumental, informational, and emotional) and consists of seven items:

1. Is there someone available to whom you can count on to listen to when you need to talk?

2. Is there someone available to you to give you good advice about a problem?

3. Is there someone available to you who shows you love and affection?

4. Is there someone available to help with daily chores?

5. Can you count on anyone to provide you with emotional support (talking over problems or helping you make a difficult decision)?

6. Do you have as much contact as you would like with someone you feel close to, someone in whom you can trust and confide in?

7. Are you currently married or living with a partner?

Items one to six could be answered with: None of the time (score 1); A little of the time (score 2); Some of the time (score 3); Most of the time (score 4); or, All of the time (score 5). The seventh item could be answered with Yes (score 4) or No (score 2) [35]. To assess instrumental and informational social support further, we added two further questions to the instrument: '*Is there someone available to you who can support you financially if needed*?' (instrumental support), and, '*Is there someone available to you who can help you read letters or flyers from, for example, the municipality, health insurance organisation, or hospital*?' (informational support). The options to answer these questions ranged from: None of the time (score 1), to All of the time (score 5). After this, we evaluated whether the questions refer to a single concept. This was studied by performing a polychoric factor analysis, since the question about having a partner is dichotomous. In addition, we examined Spearman correlations. The results of the factor analysis are presented in Table 1. As can be observed from the table, the factor analysis revealed one factor. However, the question about having a partner did not load sufficiently on this factor (factor loading = 0.40). Furthermore, Spearman correlation coefficients of the question about one's partner and the other questions ranged from 0.13 to 0.37 (see S1 Table). This indicates weak correlations [36]. Because of this, we decided not to include the partner question in the scale on perceived social support. This is in line with some other studies that have used ESSI and also found low correlations between the partner question and the other items [e.g. 37]. After removing the question about having a partner, the factor loadings and correlations of the remaining items were studied. Factor loadings of all items were sufficient (0.63 to 0.88, see Table 1), although the factor loadings of the items we added to the instrument were lower than those of the original items. In addition, Spearman correlations were moderate to strong ($r_s$ = 0.40 to 0.75, see S1 Table) [36]. We then looked at the internal consistency using Cronbach's alpha. Based on the cut-off values of George and Mallery, as cited in Gliem and Gliem [38], the internal consistency was rated excellent (Cronbach's alpha = 0.90). Because of this, all the remaining items were included in the scale on perceived social support. Next, a mean score (range 1 to 5) was calculated, with higher scores indicating higher levels of perceived social support. In the original instrument, a total score was calculated by summing up the scores on the individual items. This was necessary as the question about

**Table 1. The factor analysis for the items on social support, before and after removal of the question about having a partner.**

| | Before removal of the partner question (N = 748*) | After removal of the partner question (N = 757*) |
|---|---|---|
| | Factor 1 | Factor 1 |
| | Factor loading | Factor loading |
| 1) Is there someone available to whom you can count on to listen to when you need to talk? | 0.8646 | 0.8103 |
| 2) Is there someone available to you to give you good advice about a problem? | 0.8572 | 0.8112 |
| 3) Is there someone available to you who shows you love and affection? | 0.8382 | 0.7585 |
| 4) Is there someone available to help with daily chores? | 0.7918 | 0.7135 |
| 5) Can you count on anyone to provide you with emotional support (talking over problems or helping you make a difficult decision)? | 0.9200 | 0.8765 |
| 6) Do you have as much contact as you would like with someone you feel close to, someone in whom you can trust and confide in? | 0.8295 | 0.7764 |
| 7) Is there someone available to you who can support you financially if needed? | 0.7080 | 0.6324 |
| 8) Is there someone available to you who can help you read letters or flyers from, for example, the municipality, health insurance organisation, or hospital? | 0.7568 | 0.6416 |
| 9) Are you currently married or living with a partner? | 0.3959 | |

* Only respondents who answered all questions necessary for the factor analysis were included in the analysis.

having a partner was scored differently from the other items. However, since we did not include the partner question, we chose to calculate a mean score based on the number of questions that were filled out by the respondents. By doing this, we could include respondents, even when not all questions were answered, without having to impute missing values. For a mean score to be calculated, at least four questions had to be answered (excluded N = 62). 757 respondents answered all questions on social support. 17 respondents answered four to seven questions. In order to test whether other choices regarding the construction of the scale would have yielded different results, we also constructed a scale for which all questions had to be filled out for a score to be calculated. The mean score of this scale was 4.14, compared to 4.12 for the scale requiring four answers (see S2 Table). The results, thus, appear to be similar. Because of this, we decided to use the scale for which four questions had to be filled out, as there are fewer missing values on this scale.

**Social norms.** Regarding social norms, respondents were asked to what extent they agreed with the following statements: '*My partner thinks it is important that the costs of healthcare are paid for by society as a whole. In this way, people in good health contribute to the healthcare costs of people in poor health*'; '*My family thinks it is important that the costs of healthcare are paid for by society as a whole. In this way, people in good health contribute to the healthcare costs of people in poor health*'; and, '*The people I consider important think it is important that the costs of healthcare are paid for by society as a whole. In this way, people in good health contribute to the healthcare costs of people in poor health*'. The statements could be answered with: Completely disagree (score 1); Disagree (score 2); Not agree, not disagree (score 3); Agree

**Table 2. The factor analysis for the items on social norms (N = 589\*).**

| | Factor 1 |
| --- | --- |
| | Factor loading |
| 1) My partner thinks it is important that the costs of healthcare are paid for by society as a whole. | 0.8612 |
| 2) My family thinks it is important that the costs of healthcare are paid for by society as a whole. | 0.8821 |
| 3) The people I consider important think it is important that the costs of healthcare are paid for by society as a whole. | 0.8940 |

\* Only respondents who answered all questions necessary for the factor analysis were included in the analysis.

(score 4); or Completely agree (score 5). Respondents could also choose: 'Not applicable, I do not have..'. This option was recoded to 'missing' for all statements, since it provided no information about social norms. The statements were developed by the research team and were based upon other studies of social norms [39]. Next, we examined whether the questions refer to a single concept using factor analysis. As can be observed from Table 2, factor analysis revealed one factor on which all statements loaded sufficiently (0.86 to 0.89). Furthermore, Spearman correlations between the statements were strong ($r_s$ = 0.78 to 0.83, see S3 Table) [36] and the internal consistency was excellent (Cronbach's alpha = 0.92) [38]. Because of this, all items were included in the scale on social norms. After this, a mean score (range 1 to 5) was calculated based on the number of statements that were filled out by the respondents. A higher score indicates that the social context is more supportive of the solidarity-based healthcare system. At least one statement had to be filled out for a mean score to be calculated (excluded N = 79). 589 respondents answered all three questions. 169 respondents answered one or two questions. To assess whether a different scale construction would have yielded different results, we compared the scores on the above mentioned scale to the scores on a scale that requires all three questions to be filled out. The mean scores of both scales were similar (3.96 versus 3.95, see S4 Table). Since not all respondents had a partner, the scale based on the scores on all three questions had significantly more missing values (N = 248). Because of this, we decided to use the scale that required at least one question to be filled out.

**Demographic variables.** The demographic characteristics included in this study are: age (continuous); gender (1 = male, 2 = female); highest completed educational level (1 = low (none, primary school, or pre-vocational education), 2 = middle (secondary or vocational education), 3 = high (professional higher education or university)); and, self-reported health (1 = bad or fair, 2 = good, 3 = very good or excellent).

## Statistical analysis

Firstly, a descriptive analysis was performed to describe the characteristics of the study population. After this, we studied the relationship between the willingness to contribute to the healthcare costs of others and people's social context. To achieve this, multiple logistic regression analysis was performed using three models. The first model focused on the role of perceived social support. In this model, perceived social support was included as an independent variable. This model also included the demographic variables in order to control for any possible effects due to the composition of the sample. Model 2 examined the relationship between the willingness to contribute and social norms. This model included social norms as independent variable in addition to the demographic variables. Finally, Model 3 included the interaction between social support and social norms, as well as the demographic variables. The willingness

to contribute was included as the dependent variable in all models. The analyses were performed using Stata, version 16.1. A significance level of 5% (p = 0.05) was maintained for all analyses. Since we formulated directional, one-sided hypotheses, the p-values of the variables included in the hypotheses were divided by two if the found result was in the expected direction. For the analyses, we subtracted the observed minimum scores on the scales for perceived social support and social norms from the achieved scores, so that a value of zero corresponds to the lowest achieved score on the scale. This was done in order to be better able to interpret the results of the analyses.

# Results

## Descriptive analysis

Table 3 shows the results of the descriptive analyses. The mean age of the respondents was 57.9 years and over half were women (53%). Compared to the general population, the group of respondents consisted of slightly fewer men (50% in the general population) and fewer people in the age group 18 to 39 years (34% in the general population) [30]. Almost half of respondents (46%) had a high level of education and 40% reported their health as good. The mean

**Table 3. Descriptive statistics.**

|  | Number of respondents (N) | Percentage (%) or mean (SD) |
|---|---|---|
| **Age (range 20–91)** | **837** | 57.9 (16.6) |
| 18–39 years | 174 | 21% |
| 40–64 years | 412 | 49% |
| 65 years and older | 251 | 30% |
| Missing | 0 | 0% |
| **Gender** | **837** | |
| Male | 395 | 47% |
| Female | 442 | 53% |
| Missing | 0 | 0% |
| **Highest completed level of education** | **837** | |
| Low | 79 | 9% |
| Middle | 362 | 43% |
| High | 382 | 46% |
| Missing | 14 | 2% |
| **Perceived health status** | **837** | |
| Bad/fair | 135 | 16% |
| Good | 336 | 40% |
| Very good/excellent | 260 | 31% |
| Missing | 106 | 13% |
| **Perceived social support (range 1.125–5)** | **775** | 4.1 (0.8) |
| Missing | 62 | 7% |
| **Social norms (range 1–5)** | **758** | 4.0 (0.9) |
| Missing | 79 | 9% |
| **Willing to contribute to the healthcare costs of others** | **837** | |
| No | 169 | 20% |
| Yes | 617 | 74% |
| Missing | 51 | 6% |

score on perceived social support was 4.1 (range 1.125–5) and the mean score on social norms was 4.0 (range 1–5). Approximately three-quarters of respondents (74%) indicated that they were willing to contribute to the healthcare costs of others.

## The social context of people and their willingness to contribute to the healthcare costs of others

Table 4 and S5–S7 Tables show the relationship between people's social context and their willingness to contribute to the healthcare costs of others. First, we studied the role of perceived social support in Model 1. The likelihood ratio (LR) chi-square of this model is 24.76 (DF = 7, p = 0.001). It was found that perceived social support is significantly associated with the willingness to contribute to other people's healthcare costs. People who perceive higher levels of social support are more willing to contribute to these costs. An increase of one on the scale for perceived social support is associated with a 24% higher odds of being willing to contribute (OR = 1.24). This is in line with our expectations (hypothesis 1). Besides, Model 1 shows a statistically significant association between educational level and the willingness to contribute. People with a high level of education, as compared to people with a low one, are more willing to contribute to the healthcare costs of others (OR = 2.87). Next, we examined the role of social norms in Model 2. The LR chi-square of model 2 is 49.11 (DF = 7, p = 0.000). We observed a statistically significant association between social norms and the willingness to contribute. In accordance with hypothesis 2, we found that the willingness to contribute to other people's healthcare costs is higher when someone's social context is more supportive of healthcare systems that are financed according to the principles of solidarity. The odds of being willing to contribute is 70% higher for people scoring one point higher on the scale for social norms, all else being equal (OR = 1.70). Model 2 also shows a significant association between willingness to contribute and the demographic characteristics educational level and self-reported health. People with a high level of education and people who report their health as very good or excellent are more willing to contribute to the healthcare costs of others than those with a low level

**Table 4. The relationship between perceived social support and social norms and the willingness to contribute to the healthcare costs of others.**

| Willingness to contribute | Model 1 (N = 711) | | | Model 2 (N = 693) | | | Model 3 (N = 692) | | |
|---|---|---|---|---|---|---|---|---|---|
| | Odds ratio | SE | p-value* | Odds ratio | SE | p-value* | Odds ratio | SE | p-value* |
| **Intercept** | 0.56 | 0.36 | 0.362 | 0.19 | 0.13 | **0.012** | 0.11 | 0.14 | 0.074 |
| **Perceived social support** | 1.24 | 0.15 | **0.075** | | | | 1.22 | 0.44 | 0.583 |
| **Social norms** | | | | 1.70 | 0.18 | **0.000** | 1.69 | 0.64 | 0.166 |
| **Perceived social support * social norms** | | | | | | | 1.00 | 0.12 | 0.980 |
| **Age** | 1.01 | 0.01 | 0.150 | 1.01 | 0.01 | 0.235 | 1.01 | 0.01 | 0.174 |
| **Gender (male = ref)** | | | | | | | | | |
| Female | 0.87 | 0.16 | 0.445 | 0.94 | 0.18 | 0.748 | 0.91 | 0.18 | 0.616 |
| **Educational level (low = ref)** | | | | | | | | | |
| Middle | 1.45 | 0.42 | 0.200 | 1.66 | 0.50 | 0.091 | 1.64 | 0.49 | 0.099 |
| High | 2.87 | 0.91 | **0.001** | 2.97 | 0.97 | **0.001** | 2.90 | 0.95 | **0.001** |
| **Self-reported health (bad/fair = ref)** | | | | | | | | | |
| Good | 1.20 | 0.30 | 0.461 | 1.46 | 0.37 | 0.139 | 1.40 | 0.36 | 0.194 |
| Very good/ excellent | 1.38 | 0.38 | 0.245 | 1.76 | 0.51 | **0.048** | 1.60 | 0.47 | 0.112 |

* Statistically significant effects are in bold (p ≤ 0,05). Since we formulated directional, one-sided hypotheses, p-values were divided by two if the found result was in the expected direction. The table presents the undivided alphas.

of education (OR = 2.97) and those who report their health as bad or fair (OR = 1.76), respectively. Finally, we looked into the interaction between perceived social support and social norms in Model 3. The LR chi-square of this model is 51.06 (DF = 9, p = 0.000). We found no statistically significant interaction effect (OR = 1.00). This indicates that the association between the willingness to contribute to other people's healthcare costs and social norms does not differ between people who perceive high and low levels of social support. This does not correspond with hypothesis 3. Model 3 also shows a significant association between the willingness to contribute and educational level. People with a high level of education are more willing to contribute to the healthcare costs of others than people with a low one (OR = 2.90).

## Discussion

The aim of this study was to gain insight into the role of people's social context in order to explain differences in support for solidarity in healthcare financing. We focused on two aspects: perceived social support and social norms. With regard to perceived social support, we found that people who perceive higher levels of support are more willing to contribute to the healthcare costs of others. This suggests that the degree to which people feel connected to others affects their willingness to contribute to the welfare state. This corresponds with our expectations and is in accordance with previous research [17]. Since our study has a cross-sectional design, the causality between support for solidarity in healthcare financing and perceived social support cannot be determined. Possibly, they reinforce each other. In addition, we demonstrated that people are more willing to contribute when their social context is more supportive of healthcare systems that are financed according to the principles of solidarity. This implies that people hold views that are consistent with those of their social context. This is in line with our hypothesis and corresponds with previous studies [e.g. 19, 39, 40]. The expression of attitudes that are in line with the social context is regulated by social norms. Social norms specify which actions and attitudes are deemed appropriate—and with that, deserving of approval—and which are not [20]. In order to gain or maintain group acceptance, people will conform to the attitudes of their social context [21]. However, social norms are not the only factor contributing to people expressing views that are consistent with those from their social context. This also stems from the tendency for people to be more attracted to people who are similar to them. In other words, people are more likely to build relationships with those who share their values and beliefs in the first place. This is called selection [41, 42]. However, it is not possible to determine the causality between one's own attitudes towards the willingness to contribute and the attitudes of the social context since our study has a cross-sectional design. Further research based on longitudinal data is, therefore, recommended to study the influence of selection and social norms on the similarity of attitudes towards solidarity in healthcare financing more closely. Next to the influence of the social norms of the people in one's direct social context, views are being influenced by the broader context of social media. Social media provides the opportunity for people to communicate with others and share their views. Through these online interactions, new identities can be formed [43]. Because of this, it is recommended to look further into the role of social media in the level of support for collectively financed healthcare systems.

Our study provides no evidence for an interaction effect between perceived social support and social norms. This suggests that the relationship between social norms and the willingness to contribute does not differ between people who perceive high and low levels of social support. This is not in accordance with our hypothesis. We expected that the willingness to contribute to other people's healthcare costs would be more consistent with the norms of the social context for people who perceive lower levels of social support. Neither do our

observations correspond with previous research [e.g. 26, 44]. These past studies focused on the interaction between perceived social support and social norms among young adults. Our study not only includes young adults, but also people from other age groups. It is possible that the interaction works differently for people of different ages. Possibly, young people are more prone to peer pressure, which is why the effect of social norms may be stronger for people from this age group when they perceive low levels of social support than for older people [45]. To test this assumption, we conducted an ANOVA to study possible differences in the interaction between perceived social support and social norms between age groups. We looked at three age groups: 18 to 39-year-olds, 40 to 64-year-olds, and people aged 65 and over. No statistically significant differences by age were found. This is possibly due to the small proportion of young adults in our study. Further research into the interaction between perceived social support and social norms among different age groups is, therefore, recommended.

## Methodological considerations

Both the scales on perceived social support and social norms show a fairly high mean score. We, therefore, compared our scores to those of other studies in order to assess whether the mean scores found in this study are divergent. We found that other studies using the ESSI instrument also report high mean scores on perceived social support [e.g. 35, 37]. Moreover, our distribution of perceived social support is similar to that of other studies. Bucholz et al. [46] distinguished between low and moderate to high levels of social support using the ESSI instrument. Based on this distinction, it was demonstrated that 21% of respondents perceive a low level of social support [46]. When this distinction was applied to our study, taking into account that our scale on perceived social support deviates from the original instrument, it was found that 20% of respondents perceive a low level of social support (score of 3.6 or lower, see S8 Table). The scores on perceived social support found in this study, thus, do not seem to deviate from those demonstrated in other studies.

Regarding social norms, it is not possible to compare the scores found in this study to those of other studies. A validated instrument on social norms regarding solidarity in healthcare financing is lacking. Therefore, the research team developed its own statements to measure social norms. Since data were collected via the Dutch Healthcare Consumer Panel, it can be argued that the scores on social norms in this study are higher than those in the general population. Panel members are expected to have a more positive attitude towards the healthcare system, resulting in a possible overestimation of the norms of their social context regarding this system. This may have affected scores on social norms. But, it is not likely to have influenced the relationship between willingness to contribute and social norms since panel members are also expected to be more willing to contribute because of their positive attitude towards the healthcare system. The overestimation among panel members is expected to play a smaller role in the scores on perceived social support, since social support is broader than healthcare alone.

Social norms were measured indirectly. Panel members were asked to rate how members of their social context view healthcare systems that are financed according to the principles of solidarity, rather than questioning these social context members themselves. The assessment of panel members may deviate from the actual attitudes of their social context. However, social norms research focuses on the influence of social norms as perceived by people. Even though perceived social norms and actual norms tend to be related, they do not always correspond [47]. This justifies measuring social norms via the respondents themselves.

## Strengths and limitations

A strength of this study is that questionnaires were sent out both by post and online, thereby including people who are not skilled in digital communications. Another strength is the large sample size. A limitation of this study is that the group of respondents consisted of slightly fewer men and was on average older than the general population in the Netherlands. However, subgroup analysis showed that the relationship between support for solidarity in healthcare financing and the social context among men aged 18 to 39 years is similar to that in the rest of the population. The lower response rate in this group, therefore, does not seem to have affected our results. In addition, we looked into any possible biases due to missing data by comparing the scores and characteristics of the respondents without missings to those of the respondents with missing data. This showed a similar level of support for solidarity in healthcare financing, similar mean scores on perceived social support and social norms, and similar background characteristics. Another limitation is that although the instrument we used to assess perceived social support was validated, it was developed to measure social support among cardiology patients. In addition, we added items to the original scale in order to assess instrumental and informational social support further and translated the ESSI into Dutch. This may have affected the validity of the instrument. Lastly, the specific context of the Netherlands, where this study was conducted, affects people's support for solidarity in healthcare financing. However, we expect the mechanisms behind the relationship between the degree of support and the social context to be similar across countries. Because of this, our findings may also be insightful to countries with a different healthcare system.

## Implications

Previous studies on support for solidarity in healthcare financing have focused on mechanisms at the individual and institutional level in order to explain differences in people's levels of support. This study gives a first insight into the role of people's social context in their willingness to contribute to the healthcare costs of others. Our results suggest that the social context has to be taken into consideration in policy and research that addresses support for solidarity in healthcare financing. A possible way of doing this is through social norm nudges. Social norm nudges are behavioral interventions that inform individuals about the actions or attitudes of others. Since people are prone to follow social norms in order to avoid exclusion, it is expected that this information encourages them to act in a similar manner [48]. When applied to support for solidarity in healthcare financing, this could for instance be done by informing individuals about other people's willingness to contribute to the healthcare system.

## Conclusion

Our study demonstrates that people's social context plays a role in their willingness to contribute to the healthcare system. We showed that people who perceive higher levels of social support are more willing to contribute to the healthcare costs of others. Furthermore, the willingness to contribute is higher when the social context is more supportive of healthcare systems that are financed according to the principles of solidarity. Contrary to our expectations, the effect of social norms does not differ between people who perceive low and high levels of social support. Our results suggest that support for solidarity in healthcare financing does not only arise at the individual and institutional level. The web of social relationships surrounding an individual also plays a role.

## Supporting information

**S1 Table. Spearman correlations items social support.**
(DOCX)

**S2 Table. Results sensitivity analysis social support.**
(DOCX)

**S3 Table. Spearman correlations items social norms.**
(DOCX)

**S4 Table. Results sensitivity analysis social norms.**
(DOCX)

**S5 Table. Logistic regression analysis—Table 4, model 1.**
(DOCX)

**S6 Table. Logistic regression analysis—Table 4, model 2.**
(DOCX)

**S7 Table. Logistic regression analysis—Table 4, model 3.**
(DOCX)

**S8 Table. Distribution of scores social support instrument.**
(DOCX)

## Acknowledgments

We would like to thank all panel members for participating in the study.

## Author Contributions

**Conceptualization:** Marloes A. Meijer, Anne E. M. Brabers, Judith D. de Jong.

**Formal analysis:** Marloes A. Meijer.

**Investigation:** Marloes A. Meijer, Anne E. M. Brabers, Judith D. de Jong.

**Methodology:** Marloes A. Meijer, Anne E. M. Brabers, Judith D. de Jong.

**Project administration:** Anne E. M. Brabers, Judith D. de Jong.

**Supervision:** Anne E. M. Brabers, Judith D. de Jong.

**Writing – original draft:** Marloes A. Meijer.

**Writing – review & editing:** Anne E. M. Brabers, Judith D. de Jong.

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
