## [Decision Letter · Decision Letter 0]

9 Feb 2023

PONE-D-22-30421Social context matters: the role of social support and social norms in support for solidarity in healthcare financingPLOS ONE

Dear Dr. Meijer,

Thank you for submitting your manuscript to PLOS ONE. After careful consideration, we feel that it has merit but does not fully meet PLOS ONE’s publication criteria as it currently stands. Therefore, we invite you to submit a revised version of the manuscript that addresses the points raised during the review process.

We look forward to receiving your revised manuscript.

Kind regards,

Zsombor Zrubka, PhD

Academic Editor

PLOS ONE

Journal Requirements:

2. You indicated that ethical approval was not necessary for your study. We understand that the framework for ethical oversight requirements for studies of this type may differ depending on the setting and we would appreciate some further clarification regarding your research. Could you please provide further details on why your study is exempt from the need for approval and confirmation from your institutional review board or research ethics committee (e.g., in the form of a letter or email correspondence) that ethics review was not necessary for this study? Please include a copy of the correspondence as an ""Other"" file.

Additional Editor Comments :

Dear Authors,

the reviewers have sent their comments.

I suggest checking the reporting using an applicable guideline, such as CROSS.

I suggest to use "perceived social context" in this study.

I suggest commenting of the effect sizes of the findings.

Furthermore, I kindly ask the authors to comment on the following issues re data analysis

a)

In Model 3, when entering perceived social support and social context, neither of these variables remained significant.

Did the authors explore the reason for this?

I suggest performing a thorough model sanity check including the analysis of goodness of fit, multicollinearity etc... and comment the findings.

Maybe a model without the interaction, but including the two predictors should be presented too.

b)

I kindly ask the authors to clarify: were the minimum scores of the scale (e.g. decreasing Likert scale values by 1), or minimum scores observed in the sample subtracted from the predictor scores? Please clarify the impact of this transformation the findings.

Thank you,

Your Academic Editor

Reviewers' comments:

Reviewer's Responses to Questions

**Comments to the Author**

1. Is the manuscript technically sound, and do the data support the conclusions?

Reviewer #1: Yes

Reviewer #2: Yes

2. Has the statistical analysis been performed appropriately and rigorously? 

Reviewer #1: Yes

Reviewer #2: Yes

3. Have the authors made all data underlying the findings in their manuscript fully available?

Reviewer #1: No

Reviewer #2: No

4. Is the manuscript presented in an intelligible fashion and written in standard English?

Reviewer #1: Yes

Reviewer #2: Yes

5. Review Comments to the Author

Reviewer #1: Thank you for the opportunity to review the manuscript entitled „Social context matters: the role of social support and social norms in support for solidarity in healthcare financing”.

The authors conducted an online cross-sectional survey of the Dutch population to investigate the relationship between social context, including perceived social support and ethical norms of their social environment, and the willingness of the population to contribute to the health care costs of others. Overall, the study is written and structured following the relevant guidelines. The relevant literature has been reviewed and the objectives of the study are clearly defined. The data collection and methods used are mostly presented in sufficient detail, and the chosen analytical procedures meet the standards of the scientific field, although some additions are needed. The way in which the data are presented is adequate and the results are presented in detail, although some minor additions are necessary. The results support the conclusions of the study and the conclusions are clear. The strengths and limitations of the study are well presented and summarised, but I would nevertheless suggest a few points be added.

Overall, I find the manuscript to be a high-quality academic work, however, I have several minor comments and there are also a few methodological issues that have to be addressed.

Specific comments

Abstract

No comments.

Introduction

Rows 48-52.: The authors mention that differences in social support between certain groups have been described in the literature, that have been explained by macro and micro characteristics. However, it is not clear which groups are involved. The relevant study is indeed cited, but it would be better to explain this more clearly for ease of understanding. The reason for this is that in rows 62-66. these differences are also mentioned and it is also stated that institutional, individual, and social context characteristics are intended to be used to explain these differences.

Row 66.: „To our knowledge, this has not been studied before.”. In contrast, the abstract states that little research has been done in this area (see rows 23-24). The two sentences do not match, please clarify.

Methods

Rows 144-148: It is clear from the manuscript that according to Dutch law, neither written consent nor ethical approval is required to conduct research through the panel. It is also clear that returning the questionnaire was considered consent to participate. However, the question arises as to whether the participants were given any information at all that they were taking part in a research study, or what the purpose of the research was and what was being investigated.

Row 148: The authors describe that all methods were carried out following relevant guidelines and regulations, but none of these are mentioned in the manuscript. It would be reasonable if it were made clear which guidelines and regulations were followed for which methods.

Row 152: The questionnaire was sent to a representative sample of 1500 people. Some questions arise:

- In what way and by what methodology was representativeness ensured?

- It would be necessary to provide normal values for age and sex (either in the methods section or in the results) for either the sample of 1500 or the average Dutch population (or both), to allow for comparability.

- What considerations were taken into account when the authors determined the required sample size? (i.e. Have the authors made any sample size calculation/estimation, have they considered statistical power and expected effect size?)

Row 157:

- The response rate was quite low (56%). Do the authors have any idea what could have been the reason behind this?

- How did the low response rate affect the statistical power of the survey? (I mean, was the expected response rate anticipated and taken into account when designing the survey?)

- I was wondering whether representativeness was maintained among the respondents who returned the questionnaire since it is important for the generalisability of the results. I first found information regarding this issue in the limitations (respondents were no longer representative by age). In my opinion, this information should be briefly mentioned earlier (if we could see data for the normal population it would also be helpful, see my earlier comments).

Rows 168-169:

- The reference cited here (ref. 30) points to the ENRICHD study protocol. The ESSI scale was indeed used in this protocol, however, the actual validity study was not described in this publication. This is confusing, I suggest replacing or adding another reference (for example, ref 31 would be more appropriate).

- The ESSI is indeed a validated instrument, however, on the one hand, its validity has only been investigated in cardiology patients, and on the other hand, the original scale has been modified in the current study. While it is true that the authors have investigated the measurement properties of the scale (as described later in the methods), these analyses are not yet sufficient to fully demonstrate validity. I suggest mentioning this as part of the limitations.

- A Dutch translation of the ESSI was used in the study. It would be reasonable to briefly describe the translation process to give some insight. Furthermore, the translation itself may affect the measurement properties of the scale and thus its validity. As in the previous point, it is suggested to mention this among limitations as a problem affecting validity.

Rows 190-191:

- I recommend including the results of the Spearman correlation analysis in tabular form (as supplementary material). It is fine to include only the main results in the text, but it is also important to be able to look at all the data.

- In addition, I also suggest that for ease of interpretation of the coefficients, the cut-off values used to determine the strength of correlations (weak, medium, strong) are given.

Row 202: Also provide the cut-off values used to evaluate Cronbach's alpha.

Rows 203-205: In the relevant studies cited in the manuscript, the final ESSI score was determined by summing up the given scores for each item. However, the authors used a different method to calculate the final score (they used the mean of the scores for each item). It would be useful to draw attention to this and explain why a different calculation was used.

Rows 205-206:

- The questionnaire consisted of 8 questions in total, but it was enough to answer 4 questions for respondents to be taken into account when calculating the average. This means that there were missing data in the dataset. How exactly was the average calculated and how were the missing data handled? Was an imputation procedure applied?

- In total, how many respondents answered all the questions, and how many responded only to fewer questions (4-7 questions answered)? These data are not available in the article and should be presented.

- Perhaps it would be reasonable to briefly present the main results of the analysis mentioned in rows 346-349.

Table 1.: Maybe I missed something, but why are there only N=748 and N=757 respondents in the factor analysis? It would be helpful to address this issue in the text or add an explanation to the table as a footnote.

Rows 213-232 (Social norms): I have a few comments, which are the same as those for Perceived social support, so:

- Spearman correlations table would be useful as a supplementary material

- At least one of the three questions had to be answered by the respondents to be included in the calculation of the mean. How exactly was the mean calculated and how were missing responses taken into account?

- In total, how many respondents answered all the questions, and how many responded only to fewer questions?

- Did you perform the same sensitivity analysis as described for the other questionnaire in the methodological considerations (in rows 346-349)?

Table 2.: Why were only N=589 respondents included in the factor analysis? It would be helpful to address this issue in the text or add an explanation to the table as a footnote.

Row 240: If I understood correctly, self-reported health data was already available in the Nivel database, so it was not part of the questionnaire but was recorded at an earlier point in time. I have two questions:

- Does self-reported health measure the respondent's current health status?

- And if so, then was it taken into account that if the self-reported health data had been recorded earlier, it may have changed by the time of the survey (so not the current value was included in the analysis which may affect the results)?

Results

Table 3: It would be reasonable to indicate in the table how much data is missing for each variable (so that the table can be interpreted on its own).

Discussion, conclusions

Rows 335-336: I think this is an important and interesting observation of the study, especially in light of the previous literature. It would be good, however, if the results of previous studies and the differences from the authors’ results were presented in more detail.

Rows 346-349.: This section refers to the comments I made in the methods section on the calculation of the final score of the questionnaires. It is very good that the authors assessed how the calculated mean changes with different numbers of completed responses (this could be interpreted as a sensitivity analysis). However, I would stand by my earlier questions. It would be reasonable to present the results of this analysis in more detail (either in the methods or in the results section, or attach as a supplementary).

Row 355: ’to test’ I think is not appropriate in this context. I suggest using ’to assess’ instead

Rows 360-363: As they discuss results that were not mentioned before, it would be reasonable to include data regarding the distribution of the authors’ data as supplementary material (a table or graph showing the distribution) and to mention it among results (e.g. percentiles).

Rows 396-416: I have the feeling that the implications and conclusion paragraphs are quite similar, with many repetitive elements. I suggest modifying this, even merging it into a single paragraph.

Reviewer #2: I thank the authors for the well written research. following are some suggestions and questions:

In the introduction, the authors refer to social context as a factor to consider in solidarity assessment and proceed to investigate its impact.

The authors have chosen to investigate social norms and social support as proxies for social norms; although valid, yet they do not reflect the institutional/governmental impact on those two measures. Thinking of health as a commodity, the resulting attitudes can serve as a measure of social marketing activities, rather than social norms, as social norms might emerge as a result of targeted media campaigns. I recommend that such activities and previous literature to be discussed. examples:

Thomas, E. F., Cary, N., Smith, L. G., Spears, R., & McGarty, C. (2018). The role of social media in shaping solidarity and compassion fade: How the death of a child turned apathy into action but distress took it away. New Media & Society, 20(10), 3778-3798.

Neville, F. G., Templeton, A., Smith, J. R., & Louis, W. R. (2021). Social norms, social identities and the COVID‐19 pandemic: Theory and recommendations. Social and Personality Psychology Compass, 15(5), e12596.

some more explanation on social norms in the intro would be beneficial.

As stated in the discussion line 337 “These past studies focused on the interaction between perceived social support and social norms among young adults” . it is indeed an important conclusion. It would be great to stratify age groups and conduct a simple ANOVA to identify age group differences. Although highlighted that the number of young adults is low hindering the analysis, yet the age range is quite wide (20-90). Perhaps you can do 20-45 compared to 46-90.

Apart from the previous extra additions, i have have no comment regarding the current situation of the manuscript. the paper is well written and structured in an understandable manner.

6. PLOS authors have the option to publish the peer review history of their article (what does this mean?). If published, this will include your full peer review and any attached files.

Reviewer #1: No

Reviewer #2: **Yes: **Omar Rashdan, PhD

---

## [Author Response · Author response to Decision Letter 0]

24 Mar 2023

Comments from the editor

Journal Requirements:

Comment 1

Response to comment 1

We checked our manuscript with regard to the style requirements and made the necessary adjustments, so that the manuscript is now in line with the requirements.

Comment 2

You indicated that ethical approval was not necessary for your study. We understand that the framework for ethical oversight requirements for studies of this type may differ depending on the setting and we would appreciate some further clarification regarding your research. Could you please provide further details on why your study is exempt from the need for approval and confirmation from your institutional review board or research ethics committee (e.g., in the form of a letter or email correspondence) that ethics review was not necessary for this study? Please include a copy of the correspondence as an ""Other"" file.

Response to comment 2

As requested by the editor, we have included a copy of the correspondence with our organisation’s data protection officer on why our study does not require approval by a medical ethics committee. 

Comment 3

Please include your full ethics statement in the ‘Methods’ section of your manuscript file. In your statement, please include the full name of the IRB or ethics committee who approved or waived your study, as well as whether or not you obtained informed written or verbal consent. If consent was waived for your study, please include this information in your statement as well.

Response to comment 3

The following was added to the ethics statement in the methods section:

“Upon membership, panel members are being informed of the purpose, scope, method, and use of the panel. Based on that information, participants can give permission to participate in the panel. A written informed consent (since 2020 this consent can also be given digital) is obtained at the time of registration of a new member to the panel.” 

And

According to Dutch legislation, approval by a medical ethics committee is not obligatory for carrying out research using the panel. Participation is voluntary and members are not forced to participate in surveys, or to answer questions within the surveys. They can stop their membership at any time without having to give a reason. Respondents’ returning of the questionnaire was considered as consent to participate [29].

Comment 4

We note that you have indicated that data from this study are available upon request. PLOS only allows data to be available upon request if there are legal or ethical restrictions on sharing data publicly. For more information on unacceptable data access restrictions, please see http://journals.plos.org/plosone/s/data-availability#loc-unacceptable-data-access-restrictions. 

Response to comment 4

The minimal anonymized data set necessary to replicate our study findings has been uploaded to DANS EASY (https://doi.org/10.17026/dans-2b2-kf3a). The data set may only be used under the conditions laid down in the privacy regulations of the Dutch Healthcare Consumer Panel. Therefore, the dataset is available on request. Data requests will be assessed against the privacy regulations. Additionally, the same minimal anonymized data set is available upon request from prof. Judith D. de Jong (j.dejong@nivel.nl), project leader of the Dutch Healthcare Consumer Panel, or the secretary of this panel (consumentenpanel@nivel.nl).

Comment 5

We note that you have stated that you will provide repository information for your data at acceptance. Should your manuscript be accepted for publication, we will hold it until you provide the relevant accession numbers or DOIs necessary to access your data. If you wish to make changes to your Data Availability statement, please describe these changes in your cover letter and we will update your Data Availability statement to reflect the information you provide.

Response to comment 5

As indicated in our response to comment 4, the minimal anonymized data set necessary to replicate our study findings has been uploaded to DANS EASY (https://doi.org/10.17026/dans-2b2-kf3a). This is also mentioned in our cover letter.

Comment 6

Please include captions for your Supporting Information files at the end of your manuscript, and update any in-text citations to match accordingly. Please see our Supporting Information guidelines for more information: http://journals.plos.org/plosone/s/supporting-information. 

Response to comment 6

We included captions for our supporting information files at the end of the manuscript, as was requested by the editor.

Additional Editor Comments:

Dear Authors,

The reviewers have sent their comments.

I suggest checking the reporting using an applicable guideline, such as CROSS.

I suggest to use "perceived social context" in this study.

I suggest commenting of the effect sizes of the findings.

Response to the additional editor comments:

We carried out our study in accordance with the STROBE guideline. This was added to the methods section:

“All methods were carried out in accordance with the STROBE guideline.”

Regarding the suggestion of the editor to use ‘perceived social context’ instead of ‘social context’, we decided to keep using social context. As the phrase ‘perceived social context’ is longer, we think using this this may negatively affect the readability of our paper. In our opinion, when reading our manuscript it becomes clear that we have asked people themselves about the level of social support they experience and their estimation of the social norms of their social context. However, to emphasize that our study focuses on the social context as perceived by people, the following sentence was added to the introduction section:

“Our study focuses on the social context as perceived by people.”

Finally, to provide more insight into the effect sizes of the findings, the following sentences were added to the results section:

“An increase of one on the scale for perceived social support is associated with a 24% higher odds of being willing to contribute (OR=1.24).”

And

“The odds of being willing to contribute is 70% higher for people scoring one point higher on the scale for social norms, all else being equal (OR=1.70).”

Furthermore, I kindly ask the authors to comment on the following issues re data analysis

a)

In Model 3, when entering perceived social support and social context, neither of these variables remained significant.

Did the authors explore the reason for this?

I suggest performing a thorough model sanity check including the analysis of goodness of fit, multicollinearity etc... and comment the findings.

Maybe a model without the interaction, but including the two predictors should be presented too.

Response to comment a

Model 3 presents the results of a model in which an interaction effect between perceived social support and social context is included. Therefore, the model does not show the effects of perceived social support and social norms independently, but the effect of perceived social support if the score on the social norms instrument equals zero. And, conversely, the effect of social norms if the score on the perceived social support instrument equals zero. This may explain why no statistically significant effects were found.

To provide insight into the goodness of fit of the models, we included information on the likelihood ratio chi-square in the results section of our manuscript:

“The likelihood ratio (LR) chi-square of this model is 24.76 (DF=7, p=0.001)” , “The LR chi-square of model 2 is 49.11 (DF=7, p=0.000).” , and “The LR chi-square of this model is 51.06 (DF=9, p=0.000).”

We decided not to add a model in which both perceived social support and social norms are included, but without an interaction between the two, as such a model does not align with our hypotheses. Based on theories on the social context, we have formulated hypotheses on the effects of perceived social support and social norms independently and on the interaction between perceived social support and social norms. Since we did not formulate expectations about the effects of perceived social support and social norms when they are simultaneously included in a model, we chose not to test this. 

b)

I kindly ask the authors to clarify: were the minimum scores of the scale (e.g. decreasing Likert scale values by 1), or minimum scores observed in the sample subtracted from the predictor scores? Please clarify the impact of this transformation the findings.

Response to comment b

We subtracted the observed minimum score on the scales for perceived social support and social norms from the achieved scores. This was clarified by adding the word ‘observed’ to the following sentence:

“We subtracted the observed minimum scores on the scales for perceived social support and social norms from the achieved scores, so that a value of zero corresponds to the lowest achieved score on the scale.”

Comments from the reviewers

Reviewer 1

Thank you for the opportunity to review the manuscript entitled „Social context matters: the role of social support and social norms in support for solidarity in healthcare financing”.

The authors conducted an online cross-sectional survey of the Dutch population to investigate the relationship between social context, including perceived social support and ethical norms of their social environment, and the willingness of the population to contribute to the health care costs of others. Overall, the study is written and structured following the relevant guidelines. The relevant literature has been reviewed and the objectives of the study are clearly defined. The data collection and methods used are mostly presented in sufficient detail, and the chosen analytical procedures meet the standards of the scientific field, although some additions are needed. The way in which the data are presented is adequate and the results are presented in detail, although some minor additions are necessary. The results support the conclusions of the study and the conclusions are clear. The strengths and limitations of the study are well presented and summarised, but I would nevertheless suggest a few points be added.

Overall, I find the manuscript to be a high-quality academic work, however, I have several minor comments and there are also a few methodological issues that have to be addressed.

Abstract

No comments.

Introduction

Comment 1: 

Rows 48-52.: The authors mention that differences in social support between certain groups have been described in the literature, that have been explained by macro and micro characteristics. However, it is not clear which groups are involved. The relevant study is indeed cited, but it would be better to explain this more clearly for ease of understanding. The reason for this is that in rows 62-66. these differences are also mentioned and it is also stated that institutional, individual, and social context characteristics are intended to be used to explain these differences.

Response to comment 1:

To provide more information on the differences in support between groups that have been found in previous studies, the paragraph was changed into the following one:

“People are generally very supportive of collectively financed healthcare systems [5]. However, there are differences in the level of support between groups. Differences were found both within countries, for example by age, educational level and health status [6], and across countries [3]. Previous studies have looked into mechanisms at the individual level (micro level) and mechanisms at the level of the healthcare system as a whole (macro level) in order to explain these differences.”

We also replaced the phrase “individual and institutional characteristics” by “mechanisms at the individual and institutional level” throughout the paper, in order to make a clearer distinction between the differences found between groups and the mechanisms explaining these differences.

Comment 2:

Row 66.: „To our knowledge, this has not been studied before.”. In contrast, the abstract states that little research has been done in this area (see rows 23-24). The two sentences do not match, please clarify.

Response to comment 2:

We agree with the reviewer that the sentences mentioned above do not match. To counter this, the sentence in the introduction was changed into the following one:

“To our knowledge, little research has been conducted into this to date.”

Methods

Comment 3: 

Rows 144-148: It is clear from the manuscript that according to Dutch law, neither written consent nor ethical approval is required to conduct research through the panel. It is also clear that returning the questionnaire was considered consent to participate. However, the question arises as to whether the participants were given any information at all that they were taking part in a research study, or what the purpose of the research was and what was being investigated.

Response to comment 3:

Members of the panel are being informed about the goal of the Dutch Healthcare Consumer Panel, which is conducting scientific research through the data that are collected via the questionnaires in order to provide insight into opinions on and knowledge of healthcare among consumers, upon membership. This is also stated in the privacy policy of the panel. For every questionnaire, panel members receive information on the topic and length of the respective questionnaire via the invitation letter. To give more information about this, the following sentences were included in the paragraph on the setting in the methods section:

“Upon membership, panel members are being informed of the purpose, method, scope, and use of the panel. Based on that information, participants can give permission to participate in the panel.”

And

“Panel members were informed about the subject and length of the questionnaire via the invitation letter.”

Comment 4:

Row 148: The authors describe that all methods were carried out following relevant guidelines and regulations, but none of these are mentioned in the manuscript. It would be reasonable if it were made clear which guidelines and regulations were followed for which methods.

Response to comment 4:

We agree with the reviewer that it is relevant to make clear which guidelines and regulations were followed. Therefore, this was added to the methods section:

“All methods were carried out in accordance with the STROBE guideline.”

Comment 5:

Row 152: The questionnaire was sent to a representative sample of 1500 people. Some questions arise:

- In what way and by what methodology was representativeness ensured?

- It would be necessary to provide normal values for age and sex (either in the methods section or in the results) for either the sample of 1500 or the average Dutch population (or both), to allow for comparability.

- What considerations were taken into account when the authors determined the required sample size? (i.e. Have the authors made any sample size calculation/estimation, have they considered statistical power and expected effect size?)

Response to comment 5:

Representativeness was ensured by matching the age and gender composition of the panel members in the sample to that of the general population, using data from CBS (Statistics Netherlands). In order to do this, age was divided into three categories: 18-39 years, 40-64 years, and 65 years and older. The following sentence on representativeness was added to the Questionnaire paragraph in the methods section:

“Representativeness was ensured by matching the age and gender composition of the panel members in the sample to that in the general population, using data from Statistics Netherlands (CBS) [30].”

In addition, we reflected upon the representativeness of the group of respondents by comparing their age and gender composition to that in the general population. This was done in the results section:

“Compared to the general population, the group of respondents consisted of slightly fewer men (50% in the general population) and fewer people in the age group 18 to 39 years (34% in the general population) [30].”

Finally, no sample size calculations were made in order to determine the sample size of the study. The sample size was based on previous experiences with similar studies, which showed that when 1.500 panel members are contacted, about half of them respond. This is sufficient to perform the required subgroup analyses. Since we did not perform a randomized controlled trial, an exact sample size calculation based on power analysis was not necessary. 

Comment 6:

Row 157:

- The response rate was quite low (56%). Do the authors have any idea what could have been the reason behind this?

- How did the low response rate affect the statistical power of the survey? (I mean, was the expected response rate anticipated and taken into account when designing the survey?)

- I was wondering whether representativeness was maintained among the respondents who returned the questionnaire since it is important for the generalisability of the results. I first found information regarding this issue in the limitations (respondents were no longer representative by age). In my opinion, this information should be briefly mentioned earlier (if we could see data for the normal population it would also be helpful, see my earlier comments).

Response to comment 6:

The reviewer states that the response rate was quite low. However, we think that a response rate of 56% is reasonable. According to Story and Tait [1], a response rate of at least 40% is considered adequate for survey studies. In addition, our response rate is in line with average response rates in survey studies reported in the literature [e.g. 2; 3]. Furthermore, the response rate is similar to that of other studies conducted through the Dutch Healthcare Consumer Panel and, therefore, in line with our expectations. To provide some more information about the response rate, the following sentence was added to the Questionnaire section:

“This response rate is similar to that of other studies conducted through the Dutch Healthcare Consumer Panel. In addition, our response rate is in line with average response rates in survey studies reported in the literature [31; 32].”

References:

[1] Story, D. A., & Tait, A. R. (2019). Survey research. Anesthesiology, 130(2): 192-202. 

[2] Wu, M. J., Zhao, K., & Fils-Aime, F. (2022). Response rates of online surveys in published research: A meta-analysis. Computers in Human Behavior Reports, 100206.

[3] Holtom, B., Baruch, Y., Aguinis, H., & A Ballinger, G. (2022). Survey response rates: Trends and a validity assessment framework. Human relations, 00187267211070769.

Regarding the representativeness, we added a text in the results section in which we compare the age and gender composition of the respondents who completed the questionnaire to that of the general Dutch population (see also our response to comment 5).

Comment 7:

Rows 168-169:

- The reference cited here (ref. 30) points to the ENRICHD study protocol. The ESSI scale was indeed used in this protocol, however, the actual validity study was not described in this publication. This is confusing, I suggest replacing or adding another reference (for example, ref 31 would be more appropriate).

- The ESSI is indeed a validated instrument, however, on the one hand, its validity has only been investigated in cardiology patients, and on the other hand, the original scale has been modified in the current study. While it is true that the authors have investigated the measurement properties of the scale (as described later in the methods), these analyses are not yet sufficient to fully demonstrate validity. I suggest mentioning this as part of the limitations.

- A Dutch translation of the ESSI was used in the study. It would be reasonable to briefly describe the translation process to give some insight. Furthermore, the translation itself may affect the measurement properties of the scale and thus its validity. As in the previous point, it is suggested to mention this among limitations as a problem affecting validity.

Response to comment 7:

In accordance with the suggestion of the reviewer, we added reference 31 (Mitchell et al., 2003) to the sentence describing the ENRICHD Social Support Instrument.

With regard to the reviewer’s comments on the validity of the ESSI, we included the following in the limitations section in the discussion:

“Another limitation is that although the instrument we used to assess perceived social support was validated, it was developed to measure social support among cardiology patients. In addition, we added items to the original scale in order to assess instrumental and informational social support further and translated the ESSI into Dutch. This may have affected the validity of the instrument.”

The instrument was translated with help from a translator, who is a native English speaker. This was added to the paragraph on perceived social support in the methods section:

“Perceived social support was measured using the validated ENRICHD Social Support Instrument (ESSI), which we translated into Dutch supported by a translator who is a native English speaker.”

Comment 8:

Rows 190-191:

- I recommend including the results of the Spearman correlation analysis in tabular form (as supplementary material). It is fine to include only the main results in the text, but it is also important to be able to look at all the data.

- In addition, I also suggest that for ease of interpretation of the coefficients, the cut-off values used to determine the strength of correlations (weak, medium, strong) are given.

Response to comment 8:

In line with the suggestion of the reviewer, we added the results of the Spearman correlation analysis with the items of the ESSI to the supplementary files (see S1 Table). 

In addition, we added the source from which we derived the cut-off values that were used to interpret the results of the Spearman correlation analysis:

“Furthermore, Spearman correlation coefficients of the question about one’s partner and the other questions ranged from 0.13 to 0.37 (see S1 Table). This indicates weak correlations [34].”

Comment 9:

Row 202: Also provide the cut-off values used to evaluate Cronbach's alpha.

Response to comment 9:

As suggested by the reviewer, we have included a sentence in which we refer to the cut-off values that were used to assess the Cronbach’s alpha:

“Based on the cut-off values mentioned by George and Mallery, as cited in Gliem and Gliem [36], the internal consistency was rated excellent (Cronbach’s alpha = 0.90).”

Comment 10:

Rows 203-205: In the relevant studies cited in the manuscript, the final ESSI score was determined by summing up the given scores for each item. However, the authors used a different method to calculate the final score (they used the mean of the scores for each item). It would be useful to draw attention to this and explain why a different calculation was used.

Response to comment 10

The original score consisted of six items on which respondents could receive a score ranging from 1 to 5 and one item on which respondents could receive a score of 2 or 4 (the partner item). Because of this, it was not possible to calculate a mean score for the original scale, but instead the final score was determined by summing up the scores for each item. Since we decided not to include the partner item, as the correlation of this item with the other items was considered too weak, it was possible to calculate a mean score because all included items had similar scores. We preferred calculating a mean score over summing up the scores of the individual items, since some respondents had missing values on one our more questions. By calculating a mean score, we could determine the final score on the questions filled in by the respondents and leave out the missing values, without having to impute the data. We added the following about this in the text:

“In the original instrument, a total score was calculated by summing up the scores on the individual items. This was necessary as the question about having a partner was scored differently from the other items. However, since we did not include the partner question, we chose to calculate a mean score based on the number of questions that were filled out by the respondents. By doing this, we could include respondents, even when not all questions were answered, without imputing missing values.”

Comment 11:

Rows 205-206:

- The questionnaire consisted of 8 questions in total, but it was enough to answer 4 questions for respondents to be taken into account when calculating the average. This means that there were missing data in the dataset. How exactly was the average calculated and how were the missing data handled? Was an imputation procedure applied?

- In total, how many respondents answered all the questions, and how many responded only to fewer questions (4-7 questions answered)? These data are not available in the article and should be presented.

- Perhaps it would be reasonable to briefly present the main results of the analysis mentioned in rows 346-349.

Response to comment 11:

As is stated in our response to comment 10, the mean score is based upon the number of questions that were filled out by the respondents, with a minimum of 4. For example, the mean score of respondents who had filled in six questions was calculated from their scores on these six questions, while the mean score of respondents who had filled in eight questions was based upon eight scores. Because of this, it was not necessary to impute missing values. This is explained in the paper in the text that was added in response to comment 10.

With regard to the second point of the reviewer, the following sentences were added:

“757 respondents answered all questions on social support. 17 respondents answered four to seven questions.”

Finally, as suggested by the reviewer, we added the results of the analysis described in row 346-349:

“In order to test whether other choices regarding the construction of the scale would have yielded different results, we also constructed a scale for which all questions had to be filled out for a score to be calculated. The mean score of this scale was 4.14, compared to 4.12 for the scale requiring four answers (see S2 Table). The results, thus, appear to be similar. Because of this, we decided to use the scale for which four questions have to be filled out, as this has fewer missing values.”

Comment 12:

Table 1.: Maybe I missed something, but why are there only N=748 and N=757 respondents in the factor analysis? It would be helpful to address this issue in the text or add an explanation to the table as a footnote.

Response to comment 12:

In total, 837 panel members completed the questionnaire. However, not all respondents answered every question. In the factor analysis only respondents who answered all questions necessary for the analysis were included. To clarify this, the following sentence was added to Table 1 as a footnote: 

“Only respondents who answered all questions necessary for the factor analysis were included in the analysis.”

Comment 13:

Rows 213-232 (Social norms): I have a few comments, which are the same as those for Perceived social support, so:

- Spearman correlations table would be useful as a supplementary material

- At least one of the three questions had to be answered by the respondents to be included in the calculation of the mean. How exactly was the mean calculated and how were missing responses taken into account?

- In total, how many respondents answered all the questions, and how many responded only to fewer questions?

- Did you perform the same sensitivity analysis as described for the other questionnaire in the methodological considerations (in rows 346-349)?

Response to comment 13:

In accordance with the suggestion of the reviewer, the results of the Spearman correlations were added to the supplementary files (see S3 Table). The mean score was calculated based on the number of questions that were filled out by the respondents, with a minimum of 1. When respondents had a missing value on one or two of the questions about social norms, these questions were not taken into account when calculating their score. We clarified this by adding ‘the number of’ to the following sentence:

“After this, a mean score (range 1 to 5) was calculated based on the number of statements that were filled out by the respondents.”

With regard to the number of respondents who answered all questions versus the number of respondents who answered fewer questions, the following sentences were included:

“589 respondents answered all three questions. 169 respondents answered one or two questions.”

Finally, we did perform a sensitivity analysis similar to the one we did for the perceived social support scale. We included the results of this analysis to the method section:

“To assess whether a different scale construction would have yielded different results, we compared the scores on the above mentioned scale to the scores on a scale that requires all three questions to be filled out. The mean scores of both scales were similar (3.96 versus 3.95, see S4 Table). Since not all respondents have a partner, the scale based on the scores of all three questions had significantly more missing values (N=248). Because of this, we decided to use the scale that required at least one question to be filled out.”

Comment 14:

Table 2.: Why were only N=589 respondents included in the factor analysis? It would be helpful to address this issue in the text or add an explanation to the table as a footnote.

Response to comment 14:

Similar to the factor analysis for the items on social support, only respondents who answered all questions necessary for the factor analysis were included in the analysis. The following sentence was added to the table as a footnote:

“Only respondents who answered all questions necessary for the factor analysis were included in the analysis.”

Comment 15:

Row 240: If I understood correctly, self-reported health data was already available in the Nivel database, so it was not part of the questionnaire but was recorded at an earlier point in time. I have two questions:

- Does self-reported health measure the respondent’s current health status?

- And if so, then was it taken into account that if the self-reported health data had been recorded earlier, it may have changed by the time of the survey (so not the current value was included in the analysis which may affect the results)?

Response to comment 15:

It is true that several background characteristics of the panel members are available in the Nivel database, including self-reported health. However, in order to gain the most up-to-date insight into their health status, we again asked the respondents to assess their current general health in the questionnaire for this study. Self-reported health, thus, refers to the health status of the respondents at the time of study and has not been recorded earlier. To avoid confusion about the time of measurement, we removed self-reported health from the following sentence: “Several demographic characteristics of the panel members are known such as age, gender, educational level, and self-reported health.”

Results

Comment 16:

Table 3: It would be reasonable to indicate in the table how much data is missing for each variable (so that the table can be interpreted on its own).

Response to comment 16:

In accordance with the suggestion of the reviewer, we added information on how much data is missing for each variable in Table 3.

Discussion, conclusions

Comment 17:

Rows 335-336: I think this is an important and interesting observation of the study, especially in light of the previous literature. It would be good, however, if the results of previous studies and the differences from the authors’ results were presented in more detail.

Response to comment 17:

In row 335-336 it is described that we, in contrast to previous studies, did not find differences in the relationship between social norms and willingness to contribute between people who perceive high and low levels of social support. A possible explanation for this, as is given in the text, is that our study focuses on people from all age groups, whereas the previous studies mentioned focused on young adults only. To further elaborate on this, the following sentences were added:

“Possibly, young people are more prone to peer pressure, which is why the effect of social norms may be stronger for people from this age group when they perceive low levels of social support than for older people [43].”

Comment 18:

Rows 346-349.: This section refers to the comments I made in the methods section on the calculation of the final score of the questionnaires. It is very good that the authors assessed how the calculated mean changes with different numbers of completed responses (this could be interpreted as a sensitivity analysis). However, I would stand by my earlier questions. It would be reasonable to present the results of this analysis in more detail (either in the methods or in the results section, or attach as a supplementary).

Response to comment 18:

In line with the earlier suggestions of the reviewer, we have added more information about the results of the analyses with the scales for perceived social support and social norms for which a different number of completed questions were required. This was added in the methods section. In order to avoid repetition, we have removed the part about this topic from the methodological considerations section.

Comment 19:

Row 355: ’to test’ I think is not appropriate in this context. I suggest using ’to assess’ instead

Response to comment 19:

In accordance with the suggestion of the reviewer, ‘to test’ in row 355 was changed into ‘to assess’:

“We, therefore, compared our scores to those of other studies in order to assess whether the mean scores found in this study are divergent.”

Comment 20:

Rows 360-363: As they discuss results that were not mentioned before, it would be reasonable to include data regarding the distribution of the authors’ data as supplementary material (a table or graph showing the distribution) and to mention it among results (e.g. percentiles).

Response to comment 20:

As requested by the reviewer, we have included the data on the distribution of the scores on the perceived social support instrument as a supplementary file. We also mentioned this in our text:

“When this distinction was applied to our study, taking into account that our scale on perceived social support deviates from the original instrument, it was found that 20% of respondents perceive a low level of social support (score of 3.6 or lower, see S8 Table).”

Comment 21:

Rows 396-416: I have the feeling that the implications and conclusion paragraphs are quite similar, with many repetitive elements. I suggest modifying this, even merging it into a single paragraph.

Response to comment 21:

We agree with the reviewer that there is some overlap between the implications and conclusion sections. Because of this, we merged the paragraphs into one conclusion paragraph. 

Reviewer 2

Comment 1: 

I thank the authors for the well written research. Following are some suggestions and questions:

In the introduction, the authors refer to social context as a factor to consider in solidarity assessment and proceed to investigate its impact. The authors have chosen to investigate social norms and social support as proxies for social norms; although valid, yet they do not reflect the institutional/ governmental impact on those two measures. Thinking of health as a commodity, the resulting attitudes can serve as a measure of social marketing activities, rather than social norms, as social norms might emerge as a result of targeted media campaigns. I recommend that such activities and previous literature to be discussed. Examples:

- Thomas, E. F., Cary, N., Smith, L. G., Spears, R., & McGarty, C. (2018). The role of social media in shaping solidarity and compassion fade: How the death of a child turned apathy into action but distress took it away. New Media & Society, 20(10), 3778-3798.

- Neville, F. G., Templeton, A., Smith, J. R., & Louis, W. R. (2021). Social norms, social identities and the COVID‐19 pandemic: Theory and recommendations. Social and Personality Psychology Compass, 15(5), e12596.

Response to comment 1:

We agree with the reviewer that social media also plays a role in attitudes on solidarity-based healthcare systems. To provide more insight into this topic, the following text was added to the discussion section:

“Next to the influence of the social norms of the people in one’s direct social context, views are being influenced by the broader context of social media. Social media provides the opportunity for people to communicate with others and share their views. Through these online interactions, new identities can be formed [41]. Because of this, it is recommended to look further into the role of social media in the level of support for collectively financed healthcare systems.”

Comment 2:

Some more explanation on social norms in the intro would be beneficial.

Response to comment 2:

As suggested by the reviewer, we added the following text on social norms to the introduction section, to provide some extra information on this topic:

“Two types of social norms exist: descriptive norms and subjective norms. Descriptive norms refer to the perception of how others act in similar situations, in other words, doing wat others do. Subjective norms refer to the perception of what most people approve or disapprove of, that is, doing what others think one ought to do [19].”

Comment 3: 

As stated in the discussion line 337 “These past studies focused on the interaction between perceived social support and social norms among young adults”. It is indeed an important conclusion. It would be great to stratify age groups and conduct a simple ANOVA to identify age group differences. Although highlighted that the number of young adults is low hindering the analysis, yet the age range is quite wide (20-90). Perhaps you can do 20-45 compared to 46-90.

Response to comment 3:

In accordance with the suggestion of the reviewer, a three-way ANOVA was conducted to look into possible differences in the interaction between perceived social support and social norms between age groups. The results of this analysis are described in the discussion section:

“To test this assumption, we conducted an ANOVA to study possible differences in the interaction between perceived social support and social norms between age groups. We looked at three age groups: 18 to 39-year-olds, 40 to 64-year-olds, and people aged 65 and over. No statistically significant differences by age were found. This is possibly due to the small proportion of young adults in our study.”

Comment 4:

Apart from the previous extra additions, I have no comment regarding the current situation of the manuscript. The paper is well written and structured in an understandable manner.

Response to comment 4:

We want to thank the reviewer for this compliment.

---

## [Decision Letter · Decision Letter 1]

29 May 2023

PONE-D-22-30421R1Social context matters: the role of social support and social norms in support for solidarity in healthcare financingPLOS ONE

Dear Dr. Meijer

Thank you for submitting the revised copy of your manuscript to PLOS ONE. After due consideration, we feel that the manuscript still has merit, but requires that you address some further observations raised during the review process. Therefore, we invite you to submit a revised version of the manuscript that addresses the points raised during the review as detailed below:

Kindly address the following queries raised by one of the pee -reviewers with regards to the Limitations and Conclusions drawn in your manuscript as follows:

1. Address the issues of potential non-response bias, and generalizability of your findings in the context  of different socio-cultural milieus, as potential limitations to generalizability of the findings from your study.

2. In your discussion and conclusions, it would also be useful to address how policymakers can incorporate social context into policy design, and what specific measures could be implemented to improve solidarity in healthcare financing.

Please submit your revised manuscript by Jul 13 2023 11:59PM. If you will need more time than this to complete your revisions, please reply to this message or contact the journal office at plosone@plos.org. Please include the following items when submitting your revised manuscript:

We look forward to receiving your revised manuscript.

Kind regards,

Sylvester Chidi Chima, M.D., LL.M, LLD

Academic Editor

PLOS ONE

Journal Requirements:

Reviewers' comments:

Reviewer's Responses to Questions

**Comments to the Author**

1. If the authors have adequately addressed your comments raised in a previous round of review and you feel that this manuscript is now acceptable for publication, you may indicate that here to bypass the “Comments to the Author” section, enter your conflict of interest statement in the “Confidential to Editor” section, and submit your "Accept" recommendation.

Reviewer #1: All comments have been addressed

Reviewer #2: All comments have been addressed

2. Is the manuscript technically sound, and do the data support the conclusions?

Reviewer #1: (No Response)

Reviewer #2: Yes

3. Has the statistical analysis been performed appropriately and rigorously? 

Reviewer #1: (No Response)

Reviewer #2: Yes

4. Have the authors made all data underlying the findings in their manuscript fully available?

Reviewer #1: (No Response)

Reviewer #2: No

5. Is the manuscript presented in an intelligible fashion and written in standard English?

Reviewer #1: (No Response)

Reviewer #2: Yes

6. Review Comments to the Author

Reviewer #1: (No Response)

Reviewer #2: I congratulate the authors for the updated manuscript. I agree with the findings and I accept the paper conditional on discussing the following limitations and conclusions:

The possibility of non-response bias. It would be beneficial to investigate and compare the characteristics of respondents and non-respondents to assess the potential impact of non-response on the study results. Moreover, respondents with missing data on certain items were excluded, potentially leading to biased results.

Generalizability: The study was conducted in the Netherlands, and the findings may not be directly applicable to other countries or healthcare systems. Factors such as cultural norms, socioeconomic conditions, and healthcare system structures can significantly influence people's attitudes towards solidarity in healthcare financing. Therefore, caution should be exercised when generalizing the findings to different populations or countries.

The conclusion suggests that social support and social norms have an impact on individuals' willingness to contribute to healthcare costs. However, it is essential to consider the direction of causality. Does social support lead to a higher willingness to contribute, or do individuals with a higher willingness to contribute attract more social support?

While the conclusion suggests that social context should be considered in healthcare financing policy, it is important to critically analyze the practical implications. How can policymakers effectively incorporate social context into policy design? What specific measures can be implemented to foster a supportive social environment for solidarity in healthcare financing?

7. PLOS authors have the option to publish the peer review history of their article (what does this mean?). If published, this will include your full peer review and any attached files.

Reviewer #1: No

Reviewer #2: **Yes: **Dr. Omar Rashdan

---

## [Author Response · Author response to Decision Letter 1]

11 Jul 2023

Comments from the reviewers

Reviewer 2

I congratulate the authors for the updated manuscript. I agree with the findings and I accept the paper conditional on discussing the following limitations and conclusions:

Comment 1:

The possibility of non-response bias. It would be beneficial to investigate and compare the characteristics of respondents and non-respondents to assess the potential impact of non-response on the study results. Moreover, respondents with missing data on certain items were excluded, potentially leading to biased results.

Response to comment 1:

The questionnaire was sent to a sample of 1,500 panel members, representative of the Dutch adult population with regard to age and gender. The exact composition of the sample can be found below:

 Share in 

population Share in sample Men Women

18-39 years old 34% 510 (34%) 

 255 (50%) 255 (50%)

40-64 years old 42% 630 (42%) 

 315 (50%) 315 (50%)

65 years and older 24% 360 (24%) 

 180 (50%) 180 (50%)

Total 

100% 

1500 

750 (50%) 

750 (50%)

The questionnaire was returned by 837 panel members. The composition of the group of respondents can be found in the table below:

 Share in 

population Share in group of respondents Men Women

18-39 years old 34% 174 (21%) 

 65 (37%) 109 (63%)

40-64 years old 42% 412 (49%) 

 201 (49%) 211 (51%)

65 years and older 24% 251 (30%) 

 129 (51%) 122 (49%)

Total 

100% 

837 

395 (47%) 

442 (53%)

As can be observed from the tables above, the group of respondents consisted of fewer people in the age group 18 to 39 years (21%) compared to the general population (34%). In addition, the share of men in this age group (37%) was lower than that in the general population (50%), resulting in an underrepresentation of men in the total group of respondents (47% versus 50% in the total population). If the relationship between support for solidarity in healthcare financing and the social context among men in the age category 18 to 39 years would differ substantially from that among the rest of the population, the lower response rate in this subgroup may have led to a bias. When we looked into the relationship between support for solidarity in healthcare financing and perceived social support and social norms among men aged 18 to 39 years, we found that perceived social support is significantly associated with the willingness to contribute to other people’s healthcare costs. Men in this age category who perceive higher levels of social support are more willing to contribute to the costs of others. This is in line with the results in the total group of respondents. Contrary to the results among the total groups of respondents, we did not find a statistically significant relationship between support for solidarity in healthcare financing and social norms among men aged 18 to 39 years, although the association found was in the expected direction. Possibly, this was due to the small number of respondents in this subgroup. Therefore, we do not have a reason to assume that the relationship between support for solidarity in healthcare financing and the social context among men in the age category 18 to 39 years differs substantially from that among the rest of the population, and with that, that non-response bias has affected our results. The following text on this was included in the discussion section:

“However, subgroup analysis showed that the relationship between support for solidarity in healthcare financing and the social context among men aged 18 to 39 years is similar to that in the rest of the population. The lower response rate in this group, therefore, does not seem to have affected our results.”

In addition, we looked into the scores and characteristics of the respondents without missing data and the respondents with missing data and compared these two groups. This showed a similar level of support for solidarity in healthcare financing (79% versus 78%), similar mean scores on perceived social support (3.00 versus 2.96) and social norms (2.95 versus 3.02), and similar background characteristics. We added the following text about this to the strengths and limitations section:

“In addition, we looked into any possible biases due to missing data by comparing the scores and characteristics of the respondents without missing to those of the respondents with missing data. This showed a similar level of support for solidarity in healthcare financing, similar mean scores on perceived social support and social norms, and similar background characteristics.”

Comment 2:

Generalizability: The study was conducted in the Netherlands, and the findings may not be directly applicable to other countries or healthcare systems. Factors such as cultural norms, socioeconomic conditions, and healthcare system structures can significantly influence people's attitudes towards solidarity in healthcare financing. Therefore, caution should be exercised when generalizing the findings to different populations or countries.

Response to comment 2:

We agree with the reviewer that the specific context of the Netherlands affects people’s attitudes towards solidarity in healthcare financing. However, our study does not focus on the exact level of support for solidarity in healthcare financing in the Netherlands, but on the relationship between the degree of support and the social context. Although the level of support may vary between countries, we expect the mechanisms behind the relationship between support for solidarity in healthcare financing and the social context to be similar across countries. We added the following text about this to the strengths and limitations section:

“Lastly, the specific context of the Netherlands, where this study was conducted, affects people's support for solidarity in healthcare financing. However, we expect the mechanisms behind the relationship between the degree of support and the social context to be similar across countries. Because of this, our findings may also be insightful for countries with a different healthcare system.”

Comment 3:

The conclusion suggests that social support and social norms have an impact on individuals' willingness to contribute to healthcare costs. However, it is essential to consider the direction of causality. Does social support lead to a higher willingness to contribute, or do individuals with a higher willingness to contribute attract more social support?

Response to comment 3:

Since our study has a cross-sectional design, it is not possible to determine the causality between people’s support for solidarity in healthcare financing and their perceived level of social support. In our study, it is argued that people who feel more connected to others, because they perceive higher levels of social support, are more willing to contribute to the healthcare costs of others. This is done based on theories on perceived social support. However, as is stated by the reviewer, the willingness to contribute may also affect perceived social support. It is possible that the association between perceived social support and the willingness to contribute to other people’s healthcare costs works both ways and that these reinforce each other. The following text was added to the discussion section:

“Since our study has a cross-sectional design, the causality between support for solidarity in healthcare financing and perceived social support cannot be determined. The willingness to contribute may in turn also be affected by the level of social support. Possibly, they reinforce each other.”

Comment 4:

While the conclusion suggests that social context should be considered in healthcare financing policy, it is important to critically analyze the practical implications. How can policymakers effectively incorporate social context into policy design? What specific measures can be implemented to foster a supportive social environment for solidarity in healthcare financing?

Response to comment 4:

In accordance with the suggestion of the reviewer, we added an implications section to our paper in which we discuss the incorporation of the social context into policy design:

“Previous studies on support for solidarity in healthcare financing have focused on mechanisms at the individual and institutional level in order to explain differences in people’s levels of support. This study gives a first insight into the role of people’s social context in their willingness to contribute to the healthcare costs of others. Our results suggest that the social context has to be taken into consideration in policy and research that addresses support for solidarity in healthcare financing. A possible way of doing this is through social norm nudges. Social norm nudges are behavioral interventions that inform individuals about the actions or attitudes of others. Since people are prone to follow social norms in order to avoid exclusion, it is expected that this information encourages them to act in a similar manner [48]. When applied to support for solidarity in healthcare financing, this could for instance be done by informing individuals about other people’s willingness to contribute to the healthcare system.”

---

## [Decision Letter · Decision Letter 2]

28 Jul 2023

PONE-D-22-30421R2Social context matters: the role of social support and social norms in support for solidarity in healthcare financingPLOS ONE

Dear Dr. Meijer,

Thank you for submitting your manuscript to PLOS ONE. After careful consideration, we feel that it has merit but does not fully meet PLOS ONE’s publication criteria as it currently stands. Therefore, we invite you to submit a revised version of the manuscript that addresses the points raised during the review process.

1. After editorial review, we notice that your supporting documents (S5-S7), pertaining to the results of the Logistic regression analysis in your manuscript, are annotated in  the Dutch language. Kindly translate these Tables into English language to comply with journal requirements.2. In addition, kindly cross-check, and correct any other typographical or language errors within your revised manuscript.

Please submit your revised manuscript by Sep 11 2023 11:59PM. If you will need more time than this to complete your revisions, please reply to this message or contact the journal office at plosone@plos.org. Please include the following items when submitting your revised manuscript:A rebuttal letter that responds to each point raised by the academic editor and reviewer(s). You should upload this letter as a separate file labeled 'Response to Reviewers'.A marked-up copy of your manuscript that highlights changes made to the original version. You should upload this as a separate file labeled 'Revised Manuscript with Track Changes'.An unmarked version of your revised paper without tracked changes. You should upload this as a separate file labeled 'Manuscript'.If applicable, we recommend that you deposit your laboratory protocols in protocols.io to enhance the reproducibility of your results. Protocols.io assigns your protocol its own identifier (DOI) so that it can be cited independently in the future. For instructions see: https://journals.plos.org/plosone/s/submission-guidelines#loc-laboratory-protocols. Additionally, PLOS ONE offers an option for publishing peer-reviewed Lab Protocol articles, which describe protocols hosted on protocols.io. Read more information on sharing protocols at https://plos.org/protocols?utm_medium=editorial-email&utm_source=authorletters&utm_campaign=protocols.

We look forward to receiving your revised manuscript.

Kind regards,

Sylvester Chidi Chima, M.D., L.L.M., LLD

Academic Editor

PLOS ONE

Journal Requirements:

Reviewers' comments:

Reviewer's Responses to Questions

**Comments to the Author**

Reviewer #2: All comments have been addressed

2. Is the manuscript technically sound, and do the data support the conclusions?

Reviewer #2: Yes

3. Has the statistical analysis been performed appropriately and rigorously? 

Reviewer #2: Yes

4. Have the authors made all data underlying the findings in their manuscript fully available?

Reviewer #2: No

5. Is the manuscript presented in an intelligible fashion and written in standard English?

Reviewer #2: Yes

6. Review Comments to the Author

Reviewer #2: all comments were addressed. the authors justified all inquires effectively. congratulations to the authors. glad to accept the manuscript in its final form without any further revisions.

7. PLOS authors have the option to publish the peer review history of their article (what does this mean?). If published, this will include your full peer review and any attached files.

Reviewer #2: **Yes: **Omar Rashdan, PhD

---

## [Author Response · Author response to Decision Letter 2]

30 Aug 2023

Comments from the editor

Comment 1

After editorial review, we notice that your supporting documents (S5-S7), pertaining to the results of the Logistic regression analysis in your manuscript, are annotated in the Dutch language. Kindly translate these Tables into English language to comply with journal requirements.

Response to comment 1

In accordance with the comment of the editor, we have translated the tables mentioned into English. Our apologies, we have overlooked that these were in Dutch.

Comment 2

In addition, kindly cross-check, and correct any other typographical or language errors within your revised manuscript.

Response to comment 2

As suggested by the editor, we checked our manuscript for any typographical or language errors and have adjusted the text where necessary.

---

## [Editor Report · Decision Letter 3]

1 Sep 2023

Social context matters: the role of social support and social norms in support for solidarity in healthcare financing

PONE-D-22-30421R3

Dear Marloes Anne Meijer MSc,

We’re pleased to inform you that your manuscript has been judged scientifically suitable for publication and will be formally accepted for publication once it meets all outstanding technical requirements.

Kind regards,

Sylvester Chidi Chima, M.D., L.L.M, LLD.

Academic Editor

PLOS ONE
---

## [Editor Report · Acceptance letter]

6 Sep 2023

PONE-D-22-30421R3 

Social context matters: the role of social support and social norms in support for solidarity in healthcare financing 

Dear Dr. Meijer:

I'm pleased to inform you that your manuscript has been deemed suitable for publication in PLOS ONE. Congratulations! Your manuscript is now with our production department. 

Kind regards, 

on behalf of

Professor Sylvester Chidi Chima 

Academic Editor

PLOS ONE